# Omega-3 Fatty Acids and Their Role in Pediatric Cancer

**DOI:** 10.3390/nu13061800

**Published:** 2021-05-26

**Authors:** Alexandra Podpeskar, Roman Crazzolara, Gabriele Kropshofer, Benjamin Hetzer, Bernhard Meister, Thomas Müller, Christina Salvador

**Affiliations:** Department of Pediatrics I, Division of Hematology and Oncology, Medical University of Innsbruck, 6020 Innsbruck, Austria; Alexandra.Podpeskar@tirol-kliniken.at (A.P.); roman.crazzolara@i-med.ac.at (R.C.); Gabriele.Kropshofer@tirol-kliniken.at (G.K.); Benjamin.Hetzer@i-med.ac.at (B.H.); bernhard.meister@i-med.ac.at (B.M.); thomas.mueller@i-med.ac.at (T.M.)

**Keywords:** children, nutrition, omega-3 fatty acids, oncology, prevention, supplementation

## Abstract

Background: Malnutrition is common in children with cancer and is associated with adverse clinical outcomes. The need for supportive care is becoming ever more evident and the role of nutrition in oncology is still not sufficiently understood. In particular, the consequences of macro- and micronutrient deficiencies require further research. As epidemiological data suggest anti-tumoral properties of omega-3 (n-3) polyunsaturated fatty acids (PUFAs), we reviewed the role of nutrition and n-3 supplementation in pediatric oncology. Methods: A comprehensive literature search was conducted on PubMed through 5 February 2021 to select meta-analyses, systematic reviews, observational studies, and individual randomized controlled trials (RCTs) on macro- and micronutrient supplementation in pediatric oncology. The search strategy included the following medical subject headings (MeSH) and keywords: “childhood cancer”, “pediatric oncology”, “nutritional status”, “malnutrition”, and “omega-3-fatty-acids”. The reference lists of all relevant articles were screened to include potentially pertinent studies. Results: We summarize evidence about the importance of adequate nutrition in childhood cancer and the role of n-3 PUFAs and critically interpret findings. Possible effects of supplementation on the nutritional status and benefits during chemotherapy are discussed as well as strategies for primary and secondary prevention. Conclusion: We here describe the obvious benefits of omega-3 supplementation in childhood cancer. Further large scale clinical trials are required to verify potential anti-cancer effects of n-3 fatty acids.

## 1. Introduction

Epidemiological literature suggests a protective effect of omega-3 (n-3) polyunsaturated fatty acids (PUFAs) against cancer [1,2,3,4,5,6,7,8,9,10]. They are attributed to have significant anti-inflammatory properties, and are reported to directly inhibit carcinogenesis and tumor expansion, whilst also reducing the risk for secondary complications, thus representing a promising approach for adjunctive chemotherapy treatment [11,12,13,14].

At the same time, the incidence of malnutrition amongst children with cancer is high and both under- and overnutrition are associated with detrimental consequences, including increased risks for morbidity and mortality, early relapse rates, and a higher prevalence of secondary complications during treatment [15,16,17,18,19,20,21,22,23,24].

Taken together with the benefits of n-3 PUFA supplementation, an enhancement of the nutritional status is a potentially modifiable prognostic factor in pediatric oncology. The purpose of this review is to emphasize the importance of nutrition and the role of n-3 PUFAs as beneficial supplements in both the prevention and the treatment of pediatric cancer.

## 2. Definition and Prevalence of Malnutrition in Childhood Cancer

Malnutrition is an unspecific term used to define an inadequate nutritional status characterized by deficiencies, imbalances, or excess of energy and nutrients; it is detrimental to clinical conditions and associated with poor growth and development [15,16,17,18,19,23,24,25,26]. It is known that adequate nutrition is especially important in early childhood and is a prerequisite for the regular functioning of life-sustaining biochemical reactions, an efficient immune function, and for the general well-being of the child [10,15,16,17,18,19,27]. In malignant disease, body composition and the nutritional status are furthermore important, as they affect the response to treatment, thus subsequently resulting in different clinical outcomes [16,17,18,23,28]. Undernourished children are more often affronted by severe infectious complications; they are reported to have a reduced quality of life, and poorer neuronal and growth development [3,15,16,17,18,19,29]. On the other end of the spectrum there is also evidence for an increased risk of obesity and overnutrition that makes the children more vulnerable to long-term consequences (e.g., cardiovascular diseases or metabolic syndrome). Overweight patients are most frequently reported amongst survivors of childhood leukemia and patients treated with corticosteroids or those diagnosed with craniopharingeoma [21,28,30,31,32].

Despite the rising awareness about the importance of the nutritional status in the care of children with cancer, literature about nutritional management in patients with malignancies is scarce, and there are no routine guidelines to identify children at risk.

The incidence of undernutrition in children and adolescents with cancer varies considerably within different studies. At diagnosis, undernutrition is estimated to range between 5% and 10% in children with acute lymphoblastic leukemia to 50% in those with solid tumors including neuroblastomas and sarcomas. Children with metastatic disease are reported to be diagnosed as undernourished even more frequently [15,16,17,18,19,20]. Significant deterioration of nutritional status may also occur during and after treatment [15,16,17,18,19,25,26,28,33]. At present, most studies are retrospective and longitudinal data tracking the patients’ body compositions during the different stages of treatment are sparse. However, it seems evident that children with solid tumors are more likely to be undernourished at the time of diagnosis, whilst patients with hematological malignancies are reported to develop nutritional imbalances during and at the end of treatment. Alterations in body composition, including the loss of muscle mass (sarcopenia), have been reported particularly in children with acute lymphoblastic leukemia (ALL) [15,16,17,18,19,31,32].

Reasons for these inconclusive data regarding the prevalence of malnutrition in childhood cancer include the fact that there are different diagnostic techniques for the assessment of the nutritional status, applied for different types and at different stages of malignancy, but above all the main reason is the rather vague definition of malnutrition [15,18,19]. It must be noted that body weight is a distorting factor, as it is not a sufficiently sensitive indicator of nutritional disorders in children with cancer. Hydration levels during chemotherapy and tumor mass (e.g., embryonal neoplasms such as neuroblastoma, hepatoblastoma, or Wilms tumor) can affect body composition. Thus, children may present with normal weight and do not meet the diagnostic criteria for undernutrition despite severe deficits. Due to decreased food intake, excessive enteral losses, or enhanced requirements, undetectable nutritional depletion of macro- and micronutrients might occur in children with normal body weight. 

The lack of research in this area could be due to the fact that treating the tumor itself and the medical issues that arise in doing so are in the focus of many clinicians. A more comprehensive and holistic approach to cover individual needs and manage distinct nutritional requirements is necessary.

## 3. Etiology, Pathophysiology, and Considerable Aspects of Nutrition in Children with Cancer

The etiology and pathophysiology of malnutrition in pediatric cancer patients are multifactorial and dynamic, including iatrogenic effects due to treatment, complex metabolic disturbances, and changes in the inflammatory and hormonal system [15,16,17,18,19,34,35,36]. Whilst in some patients undernourishment is concomitant with diagnosis, a progressive deterioration of the nutritional status is frequently reported during treatment [16,17,18,19,28,36].

It is evident that malignancies alter the function of the metabolic and endocrine systems in adult and pediatric patients [15,16,17,18,19,33]. According to Selwood et al., cancer-related undernutrition is “multifactorial … and develops during malignant tumor growth, marked by changes of several hormonal and inflammatory factors yielding in an intense catabolic state” [20]. During simple starvation, the metabolism adapts to lower energy intake and gradual weight loss, aiming to preserve lean body mass at the expense of body fat. Conversely, in cancer-related undernutrition, wasting involves both the muscle and adipose tissues. Many studies also describe increased energy requirements in patients with cancer, mainly due to tumor-induced cytokine release, proteolysis, gluconeogenesis, and tumor-related muscle wasting [33,34,37,38]. However, according to the literature, only children diagnosed with solid tumors have raised resting energy expenditure (REE) rates. REE was consistently normal in children with leukemia, and, in response to treatment, energy metabolism also normalized in children with solid tumors [33]. 

In skeletal muscles and adipose tissues, cytokines and other humoral factors (i.e., Tumor-necrosis-factor-α (TNF-α), interleukin-6 (IL-6), interleukin-1-β (IL-1β), and interferon-γ (IFN-γ)) derived by the tumor, the host immune system or by mesenchymal tissues, trigger an intracellular signaling cascade, which translates into transcriptional changes in the gene-expression programs, eliciting catabolic responses [16,33,35,37,39,40,41]. Their impacts include the activation of nuclear-factor-kappa-light-chain-enhancer of activated B-cells (NF-κB), AMP-activated-protein-kinase (AMPK), and signal-transducer-and-activator-of transcription-3 (STAT3), thereby promoting inflammatory mediator transcription and worsening the anabolic processes [34,41]. It has been shown that chronic IL-6 exposure in skeletal muscle induces proteasome and autophagy protein degradation pathways that once again lead to wasting [34,35,39,40,41,42,43] (Figure 1).

Several former studies have also investigated the brain’s reactions that worsen the nutritional disturbances in response to proinflammatory cytokines [44]. Cytokines can cross the blood-brain barrier and thus trigger the secretion of hormones that affect appetite [27,36,45]. The action of n-3 PUFAs on those central nervous pathways could represent a promising therapeutic approach to treat or inhibit cancer-related weight loss [10,46]. 

Besides those tumor-induced catabolic effects, there is also a clear link between chemotherapy, its adverse effects and complications, and malnutrition. In other words, the larger the tumor mass at diagnosis, the more intensive the therapy and the higher the risk of nutritional problems [18,28]. Malnutrition, including both over- and undernutrition, is a predictor of lower tolerance for treatment and greater secondary problems, most of which again affect the nutritional status [15,16,17,18,19,32,37,47,48].

Gastrointestinal complications such as mucositis, nausea, and vomiting are amongst the main factors leading to weight loss [16,17,18,19,49]. Additionally, abdominal pain, reduced nutrient absorption, and constipation, especially in treatment regimens including vincristine, are frequently diagnosed amongst pediatric cancer patients [26,27].

Children receiving corticosteroids as part of their treatment may suffer from endocrine disturbances and experience enormous appetite fluctuations and psychological side-effects. Impaired appetite regulation triggers fat and sodium-rich food craving, stimulating the catabolic effects of glucocorticoids and building adipose tissue while deteriorating lean body mass [15]. Despite previous studies that have shown that n-3 supplementation can amend cancer-associated muscle atrophy by reducing the activity of autophagic/lysosomal and Ubiquitin-proteasome system (UPS) components and NF-kB pathway modulation [39,40,42,43], a recent animal study indicated a possible aggravation of wasting symptoms when a combination of n-3 (100 mg/kg/day) and dexamethasone (DEXA) was administered [39]. As several pediatric cancer patients undergo glucocorticoid treatment, these effects need to be further studied.

Eating and feeding problems are especially widespread among younger children. Other aggravating factors are poor pre-illness eating habits and an authoritative feeding style [27,49]. Psychological side-effects including decreased appetite, anorexia, alterations in taste and smell, and stress factors such as anticipatory nausea and vomiting are common [17,27,49]. Mental factors may again interfere with dietary intake. Caregivers report becoming less restrictive after a cancer diagnosis at the detriment of diet quality. The burden of cancer diagnosis and treatment can once again significantly alter the eating patterns of children and their families [28,49].

Malnutrition not only debilitates the body but also influences drug metabolism [11,20,23,24,25,33]. Minor changes in substance concentration may profoundly impact treatment efficacy as chemotherapeutic drugs have narrow therapeutic indices. Therefore, both over- and undernutrition can lead to severe pharmacokinetic aberrations. Undernutrition undermines drug clearance and is a determinant of treatment-increased toxicity, as frequently reported in protocols including Methotrexate (MTX) [11,47]. By contrast, a high Body Mass Index (BMI) is also associated with intensified treatment-related adverse effects. Especially in overweight children with ALL, evidence shows a higher risk of hepatotoxicity, pancreatitis, and secondary complications during premaintenance chemotherapy [30,31,32]. In obese cancer patients, changes in drug clearance, half-life, and metabolization of chemotherapeutic agents like cyclophosphamide, ifosfamide, and doxorubicin are also well known [30,36].

## 4. General Structural Features and Health Advantages of n-3 PUFAs

Polyunsaturated fatty acids can be divided into omega-3 (n-3) and omega-6 (n-6) PUFAs. Both types must be added through the alimentary canal, because the human body cannot synthesize them de novo. Foods that are naturally containing n-3 PUFAs are marine fish (herrings, halibuts, mackerels, salmons) or vegetable oils and seeds, primarily flax, canola, and soy [1,6,7,13,21]. N-3 PUFAs comprise α-linolenic acid (ALA), stearidonic acid (SDA), eicosapentaenoic acid (EPA), docosapentaenoic acid (DPA) and docosahexaenoic acid (DHA). Because PUFAs are an integral part of cell membrane phospholipids, they are decisive for membrane structure and play key roles in important signaling processes and cell-to-cell interactions [1,5,8,9,14].

Above all PUFAs are known for their impact on eicosanoid biosynthesis and function. Eicosanoids are biologically active lipids that impinge on cell growth and differentiation, immunity, inflammation, platelet aggregation, and angiogenesis [1,2,3,5,6,10,11,12,13,14,34,37]. Dietary n-3 and n-6 PUFAs are important for the metabolism of prostaglandins (PG), thromboxanes (TX), leukotrienes (LT), and hydroxyeicosatetraenoic acids (HETE) by enzymes like cyclooxygenases (COXs) and lipoxygenases (LOXs). Both types of PUFAs are sources for the production of signaling molecules; they modulate receptor signaling and gene expression. However, they have opposing mechanisms, as n-3 PUFAs are anti-inflammatory and n-6 PUFAs are pro-inflammatory [37,50]. For instance, n-3 PUFAs reduce the production of pro-inflammatory cytokines such as IL-1β, TNF-α, and IL-6 whilst simultaneously exerting indirect anti-inflammatory effects by inhibiting n-6 series eicosanoid biosynthesis. Given these reciprocal actions, higher n-6 to n-3 PUFA ratios result in an increased risk of carcinogenesis [20,48]. As chemotherapy may further exacerbate inflammation, an improvement in inflammatory profiles through n-3 PUFA supplementation could potentially increase tolerability [1,8,10,12,27].

Further known antagonistic affections of n-3 and n-6 PUFAs are described for many signaling pathways. While n-3 PUFAs downregulate the production of adhesion molecules and growth factors, n-6 PUFAs and their derivatives stimulate cellular signaling mediators including protein kinase C, ERK 1/2, Ras, and NF-κB, thereby affecting mitosis and apoptosis. By decreasing the production of basic fibroblast growth factor (bFGF) and vascular endothelial growth factor (VEGF) through suppressing cyclooxygenase-2 (COX-2) and other pro-angiogenic eicosanoids, n-3 fatty acids have been reported to inhibit tumor growth [2,48].

Their anti-cancer activity has been initially assumed in various epidemiological studies that showed that western diets, which typically exhibit poorer n-3 to n-6 PUFA ratios, were associated with higher rates of malignancies compared to traditional Asian or Mediterranean diets [1,2,3,5,8,37,48,51,52]. Both laboratory tests and animal studies proved that higher n-3 PUFA rates can be related to diminished transcription rates of specific oncogenes such as the above-mentioned Ras and AP-1 [1,31,37,48,51,52,53]. Furthermore, differences in DNA methylation have been identified between people with high and low n-3 PUFA intakes, and n-3 PUFA supplementation has been shown to induce epigenetic modifications [52,53]. Due to these diverse actions and effects on gene regulation, n-3 PUFAs have profound impact in both development and progression of cancer [5] (Figure 2).

## 5. Role of PUFAs in Pediatric Oncology

Here we will discuss evidence from the literature, mainly focusing on existing data from pediatric oncology, but also including data from some studies on adult cohorts.

It is evident that n-3 PUFAs exert their antitumoral effects by influencing multiple targets involved in different stages of cancer development, including mitosis, cell survival, angiogenesis, inflammation, and epigenetic modifications. The most important benefits are mainly ascribed to their anti-inflammatory effects that are achieved by inhibition of NF-κB and the production of suppressive mediators such as resolvins, protectins, and maresins [1,3,10,13,14]. Recently two G-protein-coupled receptors, Free Fatty Acid Receptor 1 (FFA1) and Free Fatty Acid Receptor 4 (FFA4), which have an impact on different metabolic and neurological pathways, were identified as molecular targets of n-3 PUFAs. Effects that include improving insulin sensitivity, ameliorating central nervous pathways affecting appetite, and thus regulating energy homeostasis are described [1,10,46,54]. According to recent studies, the activation of the FFA1 receptor can stimulate the release of β-endorphin, noradrenaline, and serotonin, eliciting analgesic and antidepressant effects. Additionally, this receptor could alleviate some paraneoplastic syndromes such as neuropathy, fatigue, and muscle wasting [10,54]. Concerning the possible benefits on cancer-related malnutrition, Raquel et al. reported that a higher consumption of EPA went hand in hand with lower weight loss during chemotherapy and that a higher blood DHA concentration was associated with higher BMI percentiles [10]. Rogers BC et al. described a stabilization of the REE after EPA supplementation, with significantly increased quality of life and improved appetite [2]. Concerning pediatric patients, an EPA containing nutritional supplement, which is protein and energy dense, could significantly show anti-cachectic effects in cancer patients during intensive chemotherapy [55].

N-3 PUFAs have been used to prevent carcinogenesis and inhibit malignant cell growth in vitro and in vivo [2,17,18,19,21]. As DHA is the most common FA in neural cells it is incorporated in phospholipids of the cell membranes, thus affecting membrane fluidity and receptor signaling [1,12]. Deficiencies of DHA are linked to poorer neuronal development. Furthermore, in particular neuroblastoma-, glioma-, and meningioma-cells have been reported to be profoundly deficient in DHA, whereas the levels of n-6 FA are increased [9].

Hence, Gleissman et al. examined the potential of n-3 PUFA supplementation in neuroblastoma treatment [9,56]. The mechanisms by which n-3 PUFA inhibit the progression of tumor growth have been suggested to be mediated by induction of apoptosis, via increased mitochondrial damage through intracellular accumulation of reactive oxygen species (ROS), and anti-angiogenic effects [57]. Studies also suggest that DHA may be effective in treating minimal residual disease [9]. Furthermore, Yang et al. reported that tumor cells in neuroblastoma patients did not produce as many anti-inflammatory and protective lipid mediators as normal neural cells. Thus, the cytotoxicity of chemotherapeutic agents against the tumor cells could be enhanced when used in combination with DHA, whilst normal neural cells could be protected by a reduction of oxidative stress [9,12,56].

In relation to the CNS (central nervous system), n-3 PUFA supplementation is associated with remarkable neuroprotective effects by reducing microglial activation and their impact on the Toll-like-receptor 4 (TLR4)/NFκB pathway [9]. Decreasing neuro-inflammatory processes could furthermore have an impact on psychological sequelae of malignant diseases. Evidence from studies on adult patients suggest that n-3 PUFA supplementation, possibly via the activation of FFA-4 or anti-inflammatory pathways, can improve the management of depression in cancer patients. As cytokines, such as IL-1β, TNF-α, IFN-γ can induce depression-like symptoms by reaching the hypothalamus, the downregulation of these mediators through higher n-3 PUFA ratios could represent a promising strategy for the care of psychological sequelae in pediatric patients [44,58,59,60].

In recent years, the potential benefits of n-3 PUFAs for therapy against solid and hematological tumors has also been investigated, with significant insights into the treatment of benign vascular tumors of infancy. Hemangiomas are recognized to occur secondarily to local uncontrolled angiogenesis and are known for their fast expansion. By downregulating bFGF- and VEGF-pathways and pro-angiogenic eicosanoids, such as COX-2, n-3 PUFA supplementation could represent a beneficial therapeutic option for children with vascular tumors [2].

Epidemiological and preclinical studies revealed that anticancer properties of n-3 PUFAs are attained not solely through their impact on gene regulation, but also by their modulation of pathways contributing the development of secondary complications [1,3,4,5,14,46,52]. Preclinical evidence and clinical trials in adult cancer patients showed that n-3 PUFAs might ameliorate the inflammatory response, improve nutritional status, and, consequently, quality of life of cancer patients. However, there is little data on n-3 PUFAs in pediatric cancer patients [8,12,21,25,46,48,52,53,54,61].

## 6. Role of PUFAs during Chemotherapy

Evidence from various clinical trials in adult patients proves that n-3 PUFA supplementation during treatment ameliorates tolerability of chemotherapy, regardless of the type of cytostatic drug used [4,10,12,37,50,55,60,62,63,64]. N-3 PUFAs, especially EPA and DHA, have been administered as an add-on to chemotherapy in patients suffering from different types of cancer and have been reported to improve the tumor response to treatment, protect from toxicity, and tackle secondary complications [37,46]. Several studies attribute the alleviation of symptoms like fatigue, appetite loss, nausea, vomiting, and improvement of nutritional status to n-3 PUFA supplementation [4,5,12,37,39]. Finally, the inhibitory action of D-series resolvins (RvD) on transient receptor (TRP) channels could be beneficial for cancer-related pain [62].

As explained above, n-3 PUFAs modulate the duration and the intensity of inflammatory processes, and their dietary increment could reduce promoting cytokines such as IL-1β, TNF-α and IL-6. As cytostatic therapy may exacerbate inflammation and a pro-inflammatory profile is known to be carcinogenic, supplementation of n-3 PUFAs is assumed to promote chemotherapy tolerability [10,37,63]. Several retrospective studies describe a reduction in C-reactive protein (CRP)-levels after EPA supplementation, however evidence in cancer patients is ambiguous as increased inflammatory markers could also be caused by increased cell damage [37,50]. Nonetheless, a stabilization in neutrophil/lymphocyte ratios and platelet/lymphocyte ratios as well as maintenance of neutrophil number and function could be documented [3,14]. Subsuming the activation of immune cells from both the innate and adaptive path are inhibited by ALA, DHA, and EPA, thereby reducing the risk of infections in cancer patients [17,65,66].

Another possible benefit of n-3 PUFAs during chemotherapy is their antioxidative nature. As oxidative stress is a key component in carcinogenesis and some anticancer agents such as alkylating agents and platinum-based drugs exert cytotoxicity by generating free radicals, supplementation with n-3 PUFAs represents a hopeful approach as an adjuvant therapy [12,44,66]. Studies have already demonstrated that n-3 FA ameliorated MTX-induced hepatotoxicity [11]. Suppression of ROS generation and their inactivation with antioxidants like n-3 PUFAs could be effective in treating drug-induced toxicities and improve the survival rates and quality of life of children with cancer [60,63,66].

N-3 PUFAs are instrumental in maintaining gut health and modifying the gut microbiome, reducing gastrointestinal side effects and improving the patient’s nutritional status. Dietary n-3 PUFAs are important components of enterocyte cell membranes and subsequently modify the function of the intestinal mucosal barrier. Mounting evidence suggests that the microbiome in cancer patients is altered, thus affecting inflammatory responses, absorption of nutrients, and drug metabolism. As intestinal microbes help balance the immune system, n-3 PUFA supplementation has once again been associated with relevant anti-inflammatory and antioxidative effects [48,64]. As intestinal microbial dysbiosis is an issue especially observed in patients receiving chemotherapy or radiotherapy, gastrointestinal side effects could be diminished.

Improving metabolic disturbances is another positive effect of n-3 PUFA supplementation, thereby also contributing to secondary prevention, as lipemic alterations, especially hypertriglyceridemia, are a common side effect of some cytoreductive therapies. Agents like L-asparaginase and corticosteroids are frequently used in pediatric ALL protocols and are strongly correlated with metabolic aberrations [67]. Although most patients with hypertriglyceridemia do not show any symptoms, some experience severe side effects, for example, central venous thrombosis due to increased blood viscosity [67]. We were recently able to show the positive effects of a combination therapy with n-3 PUFAs and acipimox in pediatric ALL patients to effectively and permanently normalize lipid levels within a few days without any side effects [67].

As stated above, DHA is a major component of neuronal phospholipids. Along with the recognized neuroprotective effects of n-3 PUFA derived resolvins, administration of DHA may reduce chemotherapy-related toxicities and may promote neuronal regeneration [9,60,68].

Due to their lipophilic nature and integration in the cell membranes PUFAs could also be a promising agent for tumor-targeting drug-delivery models, improving the therapeutic efficacy of chemotherapy. Therefore, PUFAs could be used as a carrier to increase the therapeutic efficacy of anticancer drugs, particularly against cancers that are resistant to treatment [12,13,29,60].

## 7. N-3 PUFAs in Primary and Secondary Prevention

As primary prevention of pediatric malignancies is difficult to achieve, research on this subject is still relatively sparse. However, commencing during pregnancy, the quantity and quality of food and additional intake of macro- and micronutrients may affect tumor genesis, and reduce the risk for comorbidities [15,16,17,18,19,27,33,52,53,55,56]. With their involvement in different mechanisms, such as cell division and apoptosis and their regulatory actions on the immune system, there are clear links suggesting that n-3 PUFAs could potentially compensate for or accentuate the effects of tumor predisposition [16,51]. Supplementation of the latter could also be a promising approach for primary prevention by impacting the oxidative status of cells and preventing the dysfunction of processes involved in carcinogenesis-like cell differentiation and apoptosis.

As children ought to grow and experience important phases of development they are more susceptible to long-term diseases following cytotoxic therapies compared to adult cancer patients [36]. Besides the immediate consequences of malnutrition in childhood cancer patients, negative effects on the course of the disease and therapy response, the effects of intensive chemotherapy on nutritional status can lead to long-term sequelae and serious problems in the later adult life. Childhood cancer survivors are reported to not only have significantly higher BMI than their peers but are also at risk for adult-onset diseases such as obesity, cardiovascular, and endocrine diseases [1,28,29,30,31,32]. As obesity is a complex syndrome and multifactorial in genesis, characterized by low-grade inflammation and deregulated lipid and glucose metabolism, nutrient dense diets including sufficient n-3 PUFA intake, appears to be a hopeful approach to restore those imbalances, thus also preventing chronic health issues and carcinogenesis [30,31,32,63].

Given the numerous benefits of n-3 PUFAs, supplementation could reduce the risk for long-term complications and facilitate secondary prevention. Consequent surveillance and regular nutritional assessments from diagnosis through treatment and long-term follow-up are required so that interventions, like n-3 PUFA supplementation, can be implemented and evaluated.

## 8. Dosing of N-3 PUFA Supplementation

Several of the aforementioned mechanisms of n-3 PUFAs suggest that a supplementation of the latter could not only be effective in the treatment of autoinflammatory diseases and cancer, but also reduce the risk for neurocognitive deficits via ensuring a proper neuronal and visual development. However, there is still limited available evidence from randomized placebo-controlled trials. Most scientific investigations focused on the effects of n-3 PUFAs on neuronal development and cognition or their supplementation in treatment regimens for neurocognitive disorders like Attention Deficit Hyperactivity Disorder (ADHD) or during pregnancy [69,70,71,72,73].

Consistent with these inconclusive results, there is no sufficient data for quantitative recommendations. The evaluated supplements varied with respect to the dose and source of the n-3 FAs DHA and EPA, ranging from 0.2 g DHA/d to 1.5 g EPA + DHA/d [6,73]. For example, the European Food Safety Authority recommends that infants and toddlers aged 6–24 months should receive 100 mg DHA/d [71]. Adults should aim to achieve a daily intake of ~250 mg DHA and EPA/d, a dose that can be attained by eating 1–2 fatty fish meal(s) per week [71]. Although there is agreement about the importance of a low n-6/n-3 ratio, optimal proportions in pediatric oncology remain to be elucidated [8,21].

## 9. Potential Adverse Effects of n-3 PUFAs

Interventional trials with n-3 PUFAs have so far not reported serious therapy-related side events. Compared to placebo, similar rates of adverse reactions are reported for n-3 PUFAs regarding short-term use, including mainly nausea and other gastrointestinal symptoms. However, PUFAs, with the nature of unsaturated bonds, are highly prone to oxidation, hence producing potentially deleterious free radicals and promoting mutagenic and carcinogenic responses. Higher doses, especially when exceeding 3 g daily, may thus have the potential to trigger inflammation and oxidative stress, whilst exerting antioxidative effects at low concentrations [6,7,73]. Other concerns include their antithrombotic actions that should be taken into consideration when used in high-risk populations, since high doses have been shown to prolong bleeding time [74]. The proposed anti-inflammatory benefits of n-3 PUFAs could possibly also decrease immune responses in already immune-compromised children, preventing appropriate inflammatory responses. However, detrimental effects to immune compromised persons have not been reported so far.

## 10. Conclusions

Malnutrition is a marginally noticed facet of cancer management in children, and supportive nutritional therapy is yet to be included into the standard care in pediatric oncology. Most cancer centers focus on treating the underlying malignancy, thereby neglecting nutritional problems. The absence of a gold standard for the evaluation of the nutritional status among children and guidelines regarding optimal nutritional intervention strategies represents a significant knowledge gap. This aspect seems even more paradoxical when considering that parents are highly motivated to integrate dietary supplements and complementary nutritional principles into their children’s care as doing so may offer a sense of involvement with the treatment and the disease process. However, most supplements have not been tested adequately to determine their safety and efficacy and their potential interactions with conventional chemotherapy, radiotherapy, or micronutrients is even less understood. Whilst some researchers propose that supplementation of antioxidants like n-3 PUFAs may lessen the toxicity of conventional therapies, there is also considerable debate about their possible harmful effects.

As n-3 PUFAs exert various beneficial effects on the immune system, various metabolism pathways and proliferation processes, administration of DHA and EPA appear to be a relatively non-toxic form of supportive therapy. However, evidence is still inconclusive and further research is required before n-3 PUFA supplementation can be recommended routinely. Although research on dietary supplements is complex, increased scientific attention is needed for the investigation of these therapies, concerning the frequent use of dietary supplements by pediatric patients with cancer. Large-scale pediatric studies on predictors of inadequate nutrition on outcome will be helpful in identifying risk factors of worse-prognosis. Randomized controlled trials on n-3 PUFAs are also particularly helpful to define the effectiveness and safety profile of n-3 PUFA supplementation in childhood cancer treatment.

## Figures and Tables

**Figure 1 nutrients-13-01800-f001:**
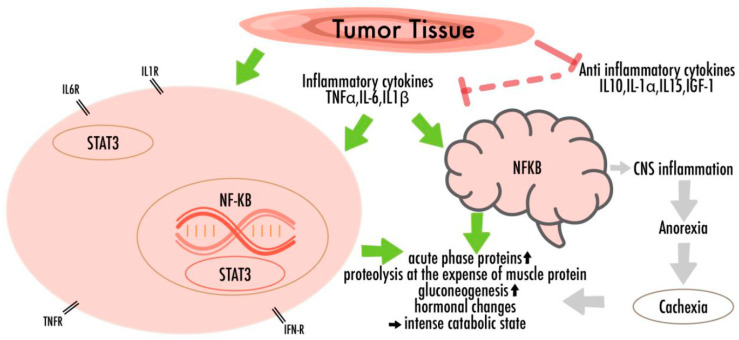
The multifactorial pathogenicity of nutritional disturbances in cancer: There are complex metabolic disturbances with systemic inflammation, negative energy and protein balance, and an unintentional loss of lean body mass. Cytokines, which lead to neuro-inflammation and increased levels of oxidative stress, are able to cross the blood-brain barrier, thus promoting behavioral consequences and appetite fluctuations.

**Figure 2 nutrients-13-01800-f002:**
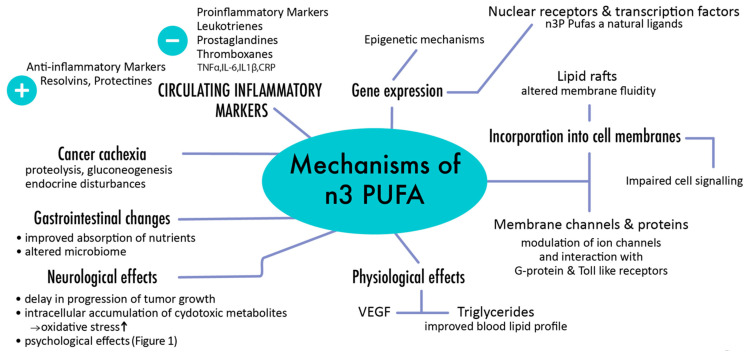
Major pathways involved in response to n-3 PUFAs.

## Data Availability

Publicly available literature was analyzed in this review. This data can be found on PubMed.

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
