# Peer review of "Omega-3 Fatty Acids and Their Role in Pediatric Cancer"

_nutrients, 2021, doi:10.3390/nu13061800_

Round 1
Reviewer 1 Report
This is a high-end study examining Omega-3 PUFA's use in children. I do have the following remarks, though:
- Too high a dose of Omega-3 PUFA's can be noxious, and the authors should add a small paragraph concerning this. Is there a recommended dose (range) to be taken?
- In adults, the Omega-6/Omega-3 PUFA relative ratio seems more important than the absolute doses of both. As children, however, may suffer from cancer types usually not seen in adults, the authors should note that extrapolation from adults to children may be hazardous.
- The authors should state why they did not conduct a meta-analysis on the (although limited) clinical part of the data.
Reviewer 2 Report
Comprehensive review of 3 Omega fatty acids biological actions and potential use in cancer therapy and modifying chemotherapy side effects.
Some deficiencies in the discussion
- While alluding to gene regulation there should be more discussion on the potential for epigenetic action
- There is always concern that adding supplements may affect chemotherapy efficacy as has been noted with antioxidants. This concern should be addressed.
- Micronutrients are numerous and other bio active foods are even more numerous. One micronutrient as discussed in this article is likely dependent on its interactions with other micronutrients or bio active foods. This aspect of the complexity of nutrition should also be addressed.
- Is there potential for toxicity from excess consumption.
- Need some discussion on Obesity as a modifiable prognostic factor and whether 3 Omega fatty acids has any role in this observation.
- Recommendation on how a clinical study could be undertaken in paediatric oncology
Reviewer 3 Report
Review comments
1220566-peer-review-v1
Omega-3 fatty acids and their role in pediatric cancer
Overall comments:
Overall the topic is an important contribution to the literature; however, it does not have a clear direction and message. The title states that the topic is on omega-3 fatty acids and their role in paediatric cancer. But when reading the article, there is only a small section that focuses on this and much of the literature and narrative are disjointed.
I suggest that the authors focus on two aspects, the role of w-3 fatty acids intake on clinical and nutritional outcomes of paediatric cancer patients during treatment and in survivorship.
The authors should clearly state when referring to adults or paediatric cancer throughout and at what stage of the cancer journey they are referring to i.e. is it prevention, during intensive treatment, post-treatment or in survivorship?
Please reference/cite appropriately throughout the manuscript
Specific comments:
Introduction:
Line 39 states that the aim of this review is to “highlight the role of n-3 PUFAs in cancer prevention and the treatment of pediatric cancer”. There is no evidence to show that nutrition intake has a role on the development and/or diagnosis of paediatric cancer diagnoses. In my view the authors should focus on the use of w-3 fatty acids during treatment and post-treatment and its impact on clinical and nutritional outcomes. The role of w-3 fatty acids on secondary cancer in survivors of childhood cancer is more appropriate.
Line 52 and throughout the article the authors use the term micronutrients to describe w-3 fatty acids. Please change the word micronutrients for macronutrients as ω-polyunsaturated fatty acids are classified as macronutrients (fats)
Line 52. The authors state that “Despite the rising awareness of cancer cachexia and its consequences, little is known about the prevalence of micronutrient deficiency in patients with malignancies”. This statement has not been referenced and the authors should be cautious about the use of the term cachexia when referring to undernutrition in paediatric cancer patients. Unlike adults, children diagnosed and treated for cancer with a curative intent rarely experienced cachexia. Furthermore, both, undernourished and over nourished children experience micronutrient (or other nutrient deficiencies). I would advise the authors to revisit this section and address the points highlighted here.
Lines 55 – 75. Definitions and prevalence of malnutrition in childhood cancer:
Please add data from key papers:
- Iniesta RR, et al. (2015) Effects of pediatric cancer and its treatment on nutritional status: a systematic review. doi: 10.1093/nutrit/nuu062. Epub 2015 Mar 29.
- Revuelta Iniesta, et al. (2019) Nutritional status of children and adolescents with cancer in Scotland: A prospective cohort study. DOI: https://doi.org/10.1016/j.clnesp.2019.04.006
- Brinksma et al. (2015). Changes in nutritional status in childhood cancer patients: a prospective cohort study. doi: 10.1016/j.clnu.2014.01.013
- Iniesta RR, et al (2021) Micronutrient status influences clinical outcomes of paediatric cancer patients during treatment: A prospective cohort study DOI:https://doi.org/10.1016/j.clnu.2021.03.020
The authors should also link this information with main theme w-3 fatty acids and their role in paediatric cancer.
Line 79. Please use undernutrition when referring to low weight, poor growth, etc. and malnutrition when referring to both under and overnutrition (overweight and obesity)
Line 83. How does malignancies in paediatric cancer affect hormonal and metabolic changes? And what is the impact of this on short and long term clinical outcomes? The information provided here is too general and does not necessarily apply to this population
Lines 127 – 124. The authors should mention the forms of nutritional support that patients receive as a results of these effects and critically appraise the evidence.
Lines 154 – 168 or lines 189 - 251. What foods are rich in w-3 fatty acids? How much w-3 fatty acids do paediatric cancer patients and survivors consume (if known)? From the nutritional interventions available in clinical practice, how much/what percentage of w-3 fatty acids in supplemented? Please consider this information.
Line 255: “Evidence from several clinical trials proves that n-3 PUFA supplementation during cancer chemotherapy ameliorates chemotherapy tolerability, regardless of the type of chemotherapy used”. In what population?
Line 305. “Nutrition as prevention of disease” subheading. This subheading is misleading as nutrition refers to all nutrition intake (macronutrients and micronutrients, food groups, etc). Furthermore, “nutrition as prevention of disease” is too general. If the authors plan to discuss the role of w-3 fatty acids in the prevention of treatment induced side-effects (short and long term) that’s what they should call it. However, this title would be another article in full by itself.
Reviewer 4 Report
This is an interesting comprehensive review article mainly focusing on the role played by omega-3 fatty acids in pediatric cancers. Below some comments that may help to improve the manuscript:
- Paragraph "Etiology, pathophysiology and important aspects of nutrition in children with cancer": when referring to humoral factors which drive cancer cachexia, IL6 should be mentioned. This is particularly relevant since this interleukin is also represented in figure 1. In addition, the authors reported that pro-cachectic factors induce muscle atrophy during cancer cachexia by mediating the activation of NFkB and STAT3. This is obviously true, however other important mechanisms have been recently elucidated. For instance, it has been reported that IL6 may activate AMPK/FoxO3 axis in skeletal muscle, which strongly contributes to muscle catabolism observe in cachectic syndrome (https://doi.org/10.1038/s41467-017-01645-7) This information should be added to render more complete the description of the molecular mechanisms. Coherently, this new detail should also be inserted in the figure 1.
- Despite omega-3 are generally reported as beneficial molecules for muscle physiology, other data indicate that omega-3 supplementation may worsen muscle atrophy induced by corticosteroids (doi: 10.14814/phy2.13966). These results should be discussed in the manuscript, especially because, as already stated by the authors, several pediatric cancer patients undergo glucocorticoid treatment.
- The authors are encouraged to insert a table summarizing the studies showing the outcomes of omega-3 supplementation in pediatric cancers.
Round 2
Reviewer 4 Report
The authors properly addressed the reviewer's concerns.